# Snapshot of Phenotypic and Molecular Virulence and Resistance Profiles in Multidrug-Resistant Strains Isolated in a Tertiary Hospital in Romania

**DOI:** 10.3390/pathogens12040609

**Published:** 2023-04-17

**Authors:** Bianca Simona Truşcă, Irina Gheorghe-Barbu, Marina Manea, Elvira Ianculescu, Ilda Czobor Barbu, Luminița Gabriela Măruțescu, Lia-Mara Dițu, Mariana-Carmen Chifiriuc, Veronica Lazăr

**Affiliations:** 1Fundeni Clinical Institute, 022328 Bucharest, Romania; 2Department of Microbiology and Immunology, Faculty of Biology, University of Bucharest, 060101 Bucharest, Romania; 3Research Institute of the University of Bucharest–ICUB, 91-95 Spl. Independentei, 050567 Bucharest, Romania; 4Fundeni Clinical Institute, University of Medicine and Pharmacy “Carol Davila” Bucharest, 020021 Bucharest, Romania; 5Romanian Academy, 050045 Bucharest, Romania

**Keywords:** multidrug resistant Gram-negative bacilli, antibiotic resistance, virulence genes, pathogenicity, bacterial fitness, healthcare associated infections

## Abstract

A current major healthcare problem is represented by antibiotic resistance, mainly due to multidrug resistant (MDR) Gram negative bacilli (GNB), because of their extended spread both in hospital facilities and in the community’s environment. The aim of this study was to investigate the virulence traits of *Klebsiella pneumoniae*, *Acinetobacter baumannii,* and *Pseudomonas aeruginosa* MDR, XDR, and PDR strains isolated from various hospitalized patients. These GNB strains were investigated for the presence of soluble virulence factors (VF), such as hemolysins, lecithinase, amylase, lipase, caseinase, gelatinase, and esculin hydrolysis, as well as for the presence of virulence genes encoding for VF involved in adherence (*TC*, *fimH*, and *fimA*), biofilm formation (*algD*, *ecpRAB*, *mrkA*, *mrkD*, *ompA*, and *epsA*), tissue destruction (*plcH* and *plcN*), and in toxin production (*cnfI*, *hlyA*, *hlyD*, and *exo* complex). All *P. aeruginosa* strains produced hemolysins; 90% produced lecithinase; and 80% harbored *algD, plcH,* and *plcN* genes. The esculin hydrolysis was detected in 96.1% of the *K. pneumoniae* strains, whereas 86% of them were positive for the *mrkA* gene. All of the *A. baumannii* strains produced lecithinase and 80% presented the *ompA* gene. A significant association was found between the number of VF and the XDR strains, regardless of the isolation sources. This study opens new research perspectives related to bacterial fitness and pathogenicity, and it provides new insights regarding the connection between biofilm formation, other virulence factors, and antibiotic resistance.

## 1. Introduction

The ability of a bacterial strain to both colonize the host organism and trigger a disease is related to its pathogenic potential, which is enhanced by the expression of one or more virulence factors [1,2]. The clinical presentation of an infectious disease depends on these major properties of a pathogen (pathogenicity and virulence), but also on several host-related factors (such as age, immune status, diet, and other environmental conditions). Different aggressins (enzymes, invasins, toxins, and secretion systems), adhesins (pili, fimbriae, and EPS components), siderophores, and the generation of biofilms are often included among the most studied virulence factors of the potentially pathogenic bacteria. Some virulence genes are chromosomally encoded, i.e., some adhesins, but others are located in plasmids or in other mobile genetic elements. Therefore, most of the AR (antibiotic resistance) and virulence genes are often co-located and could be transmitted horizontally [3]. Both have co-evolved, but with different rhythms and consequences overtime [4]. AR is an ancient evolutionary phenomenon that has been recently accelerated by various factors involved in the process of natural selection. Antibiotics are competitive factors with a naturally production rate in microbial metabolism. However, during the last eight decades, after the increased availability of different antibiotics, AR has become a global threat. Antibiotics certainly contributed to the worldwide reduction in morbidity and mortality rates due to infectious diseases, but their overuse or misuse effects, especially for the cheaper ones, led to a paradoxical effect and a continuously increasing AR [5]. On the other hand, the AR genes are not only a survival advantage for pathogens, but they also contribute to an enhancement of the virulence and the fitness of multidrug resistant (MDR) strains [6].

Some authors emphasize that, in order to improve the management of healthcare associated infections (HAIs), the detection of the virulence genes in MDR GNB is as equally important as the phenotypic and the genotypic description of AR [7]. The newly developed differences in the permeability of the outer membrane (OM) of Gram-negative bacteria has led to the initiation and extension of the AR phenomenon. The modified efflux pumps and the proteins of OM are major contributors to the intrinsic antibiotic resistance mechanisms in GNB [8]. However, the activity of the efflux pumps and OM proteins are associated with several virulence factors in GNB, according to many large protein databases (UniProt; VFDB) [9]. Resistance to β-lactam antibiotics is conferred by the production of antibiotic-modifying enzymes, efflux pumps, porins, protection of the antibiotic target, or biofilm production [10,11,12]. However, the primary mechanism of β-lactam resistance in GNB is represented by the production of β-lactamases (serine-β-lactamases: Ambler classes A, C, and D, and metallo-β-lactamases or MBL: Ambler class B) [13].

*Pseudomonas aeruginosa* (*P. aeruginosa*) is one of the most clinically relevant pathogens responsible for a high percentage of MDR infections in immunodeficient patients. The major concern regarding this microorganism is its great adaptability, which is correlated with many virulence factors and virulence-associated proteins. Their role over resistance mechanisms is currently being investigated, mostly because there is an increased need to develope new strategies for preventing and managing AR. The biofilm formation and quorum sensing processes are among the focus points, together with the functionality of the T3SS components and the secreted bacterial toxins [14]. Recent studies concerning biofilm forming bacteria have suggested the associative effect over the mechanisms of resistance for both the inflammatory endotoxins/lipopolysaccharides and the expression of the stress genes [15,16]. According to some authors, the quorum sensing mechanism and signal molecules are associated with biofilm formation and intrinsic resistance mechanisms in *P. aeruginosa.* The environment where the microorganism lives also plays a major role in the activation and functionality of the virulence factors [17].

Regarding *Klebsiella pneumoniae* (*K. pneumoniae*) strains, colonization is influenced by the presence of several virulence factors such as pili (especially coded by *mrk* and *fim* genes), efflux pumps, lipopolysaccharides, type VI secretion system, capsule, and siderophores. However, the ability of the MDR strains carrying virulence genes to produce infection after they colonize the human host remains an intensely debated subject [18]. Some authors underline the co-occurrence of a virulence plasmid and KPC genes in hypervirulent strains of *K. pneumoniae* isolated from patients with bacteriemia [19].

Among the non-fermentative GNB, *Acinetobacter baumannii* (*A. baumannii*) is also studied regarding its quorum sensing mechanisms, which are related to both the activation and the release of the virulence genes, as well as multidrug resistance. The goals are to identify the best drugs that target such mechanisms, and to reduce the excessive antibiotic burden, which eventually leads to AR [20]. This growing need to reduce multidrug resistance is a major target for microorganisms such as *A. baumannii,* which have various ways of avoiding the action of antibiotics. A series of studies have emphasized the presence of multiple resistance genes associated with this microorganism, which now encompasses almost all antibiotic classes [21]. This peculiarity represents a major epidemiological issue, especially in the era of the emergence of new viruses, such as COVID-19, which may infect the human host together with a bacterium. The attention of the researchers falls once again over a series of virulence associated factors, such as *OmpA*, which are often connected to β-lactam resistance [21].

An important epidemiological feature of these three opportunistic pathogens, which are members of the ESKAPE group, is their ability to colonize and then to cause an infection in hospitalized patients. This process is due to both the presence of resistance and to virulence factors synthesized in a manner dependent on the cellular density, as well as the mechanism of quorum sensing (QS), which protects bacterial cells (favoring their multiplication and host colonization) against antimicrobial substances that are part of the natural non-specific defense mechanisms of the host [22,23].

The aim of our study was to investigate the profiles of both the resistance and virulence of some selected MDR strains belonging to clinically significant opportunistic pathogens such as *Klebsiella pneumoniae, Acinetobacter baumannii,* and *Pseudomonas aeruginosa,* often involved in healthcare-associated infections.

## 2. Materials and Methods

### 2.1. Bacterial Strains

Bacterial strains from various biological samples (urine, blood, bile, wound, bronchial aspirate, rectal swab, drainage tube, ascitic fluid, and stool) of hospitalized patients were isolated from units with a high epidemiological risk from the Fundeni Clinical Institute during October 2019–October 2020. All of the strains were identified by an automatic method (BD Phoenix Vitek 2 Compact) and further investigated for antibiotic resistance [24]. The multidrug-resistance profile was established according to the international guidelines [25]. Furthermore, the expression of several virulence factors was investigated in a selected set of 72 MDR, XDR, and PDR strains of *K. pneumoniae* (n = 51), *A. baumannii* (n = 10), *A. lwoffii* (n = 1), and *P. aeruginosa* (n = 10) with phenotypic and molecular methods. *Escherichia coli* ATCC 25922 and *P. aeruginosa* ATCC 27853 were used as the reference strains.

### 2.2. Evaluation of Antibiotic Susceptibility

The antibiotic susceptibility testing was investigated according to CLSI 2021 (Clinical Laboratory Standard Institute), using disc diffusion and automatic methods (BD Phoenix/Vitek 2 Compact). Phenotypic confirmation for ESBL production was performed through a double-disc synergy test (DDST) (Ceftazidime–Amoxicillin/Clavulanic acid–Ceftriaxone) [26].

### 2.3. Detection of β-Lactamase Encoding Genes 

DNA extraction was performed by an adapted alkaline extraction method. Briefly, 1–5 bacterial colonies from pure cultures were suspended in 1.5 mL tubes with 20 μL NaOH 0.05 M (sodium hydroxide) and SDS 0.25% (sodium dodecyl sulphate), heated at 95 °C for 15 min followed by the addition of 180 μL TE buffer (TRIS + EDTA) 1×, centrifugation at 13,000 rpm for 3 min, and lastly the supernatant was stored at 4 °C. Simplex and multiplex PCR (PCR thermal BioRad thermocycler) amplifications were performed for *bla*_NDM_, *bla*_OXA-48_, *bla*_KPC_, *bla*_VIM_, *bla*_IMP_, *bla*_SHV_, *bla*_TEM_, *bla*_CTX-M_, *bla*_OXA-23_, *bla*_OXA-24_, and *bla*_OXA-51_ genes, using specific primers (Table 1) in a final volume of 20 µL (PCR Master Mix 2×, Thermo Fisher Scientific, Bucharest, Romania), with a content of 1 µL of bacterial DNA, after using the following conditions: initial denaturation (95 °C, 10 min), followed by 30 cycles of denaturation (95 °C, 30 s), alignment of primers (52 °C, 40 s), extension (72 °C, 50 s), and final extension (72 °C, 10 min). DNA fragments were analyzed by electrophoresis in a 1% agarose gel, at 100 V for 1 h in TAE 1× (40 mmol/L Tris-HCl [pH 8.3], 2 mmol/L acetate, 1 mmol/L EDTA) containing 0.05 mg/L SYBR Safe DNA (Thermo Fisher Scientific, Bucharest, Romania). Positive strains for *bla*_OXA23,-24,-51_ genes (in *A. baumannii*); *bla*_NDM_, *bla*_OXA-48_, *bla*_SHV_, *bla*_TEM_, and *bla*_CTX-M_ genes (in *K. pneumoniae*); and for *bla*_VIM_, and *bla*_IMP_ genes (in *P. aeruginosa*), from the Microbiology Collection of the Research Institute of the University of Bucharest, were used as the positive controls.

### 2.4. Evaluation of the Soluble Enzymatic Factors

The virulence phenotypes were assessed by performing enzymatic tests for the expression of some soluble virulence factors, as described elsewhere [27]. Briefly, soluble virulence factors were detected using solid culture media supplemented with 5% sheep blood, 2.5% yolk, 1% Tween 80, 15% soluble casein, 3% gelatine, 1% starch and Fe3+ citrate for the bacterial toxins/enzymes.

### 2.5. Virulence Genes Detection

The genetic background of virulence was investigated by PCR for selected genes from UniProt and VFDB databases and is included in Table 1, along with the genes involved in the biofilm formation (*algD*, *ecpRAB*, *mrkA*, *mrkD*, *ompA*, and *epsA*), adherence (*TC*, *fimH*, *fimA*), tissue distruction (*plcH* and *plcN*), and toxin production (*cnfI*, *hlyA*, *hlyD*, and *exo* complex). Simplex and multiplex PCR reactions were conducted using a reaction mix of 20 μL (PCR Master Mix 2×, Thermo Scientific) with 1 μL of bacterial DNA obtained using the adapted alkaline extraction method, with the following conditions: 1 cycle at 95 °C, 2 min; followed by 30 cycles at 94 °C for 30 s; 42, 55, 57, 58, 59, 60, 61, 62, and 65 °C (*cnf1*/*algD*/*ecpRAB*/*fimA*/*ompA*, *epsA*, *fimH*/, *exoT*/*exoU*/*TC*/*hlyA*, *mrkA*, and *mrkD*/*exoS*) for 30 s, 72 °C for 1 min, and a final cycle at 72 °C for 7 min. The amplification products were visualized by electrophoresis on with same parameters described above. Positive strains for *epsA* and *ompA* genes (in *A. baumannii*); *fimH*, *fimA*, *hlyD*, *hlyA*, *cnf1*, *ecpRAB*, *mrkA,* and *mrkD* genes (in *K. pneumoniae*); and *plcH*, *plcN*, *exoU*, *exoT*, *exoS*, *algD*, and *TC* genes (in *P. aeruginosa*) from the Microbiology Collection of the Research Institute of the University of Bucharest were used as the positive controls.

The primers used during the PCR reaction are depicted in Table 1.

### 2.6. Statistical Analysis

The statistical analysis was performed with the programs Microsoft Excel 2021, Vassarstats [28], and PSPP 1.6.2 [29]. The obtained frequency values were compared using Fisher Exact Probability Test and chi-square test. The correlations depicted below were established using Pearson’ s correlation coefficient in bivariate correlation and a *p* value less than 0.05 was considered statistically significant.

## 3. Results

A total of 72 bacterial strains of *K. pneumoniae* (51), *A. baumannii* (10), *A. lwoffii* (1), and *P. aeruginosa* (10) were isolated from clinical samples during October 2019 and October 2020 (Table 2).

The tested strains of *K. pneumoniae*, *P. aeruginosa,* and *Acinetobacter* spp. showed high rates of resistance to all of the tested antibiotics (Appendix A). All of the *K. pneumoniae* strains were resistant to fluoroquinolones (ciprofloxacin and/or levofloxacin), piperacillin/tazobactam, carbapenems (ertapenem and/or imipenem and/or meropenem), and third-generation cephalosporins (ceftriaxone and cefotaxime); 74.5% were resistant to trimethoprim/sulfamethoxazole; 45.1% of strains were resistant to gentamycin; 29.4% showed resistance to amikacin; and only 17.6% were resistant to ceftazidime/avibactam. The *P. aeruginosa* strains were resistant to aminoglycosides, fluoroquinolones, ceftazidime, carbapenems (imipenem and meropenem), while 30% of them displayed resistance to piperacillin/tazobactam. All *P. aeruginosa* strains were susceptible to ceftazidime/avibactam. The *A. baumannii* strains were resistant to aminoglycosides, fluoroquinolones, ceftazidime, and carbapenems (imipenem and meropenem), 54.5% were resistant to trimethoprim/sulfamethoxazole, while a lower percent (18.2%) displayed a resistance pattern to ampicillin–sulbactam and minocycline.

The *K. pneumoniae* strains carried ESBLs and carbapenemase encoding genes of *bla*_CTX-M_ (49%, 25/51), *bla*_SHV_ (45.1%, 23/51), *bla*_TEM_ (37.3%, 19/51), *bla*_KPC_ (33.3%, 17/51), *bla*_OXA-48_ (31.4%, 16/51), and *bla*_NDM_ (17.6%, 9/51) (Appendix A). In *A. baumannii,* two class D carbapenemase encoding genes were present (*bla*_OXA-23_ in 54.5% of the strains and *bla*_OXA-24_ in 9.1%) (Appendix A).

The phenotypic virulence analysis of *P. aeruginosa* revealed the production of haemolysins (100% of the strains), while 90% of the strains isolated from the blood cultures produced lecithinase. 

The majority (96.1%) of *K. pneumoniae* strains hydrolysed esculin producing esculetol, which acted as a pseudo-siderophore, and 25.5% were amylase producers. Only 11.8% of the *K. pneumoniae* strains were lecithinase producers, while gelatinase was found in 9.8% of them. Here, 31.4% of the *K. pneumoniae* strains produced lipase. We included, in Table 3, an analysis regarding the probability of encountering some of these VF in several isolation sources. Only caseinase in *K. pneumoniae* strains was more likely to be source-dependent (*p* = 0.047).

In Appendix A, we show the virulence and resistance profiles of the identified strains of *Acinetobacter* spp., *P. aeruginosa,* and *K. pneumoniae*. No significant association was encountered between the virulence or resistance profiles and the isolation sources. 

Next, we searched for an association between a certain virulence pattern and the isolation source. The association of *fimA* and *mrkA* genes was more likely to have a source-related distribution (*p* = 0.03) (see Table 4).

Because most of the analyzed strains were *K. pneumoniae*, we continued our correlation analysis on them. Thus, after indicating the source of isolation (urine, blood culture and other sources), we used a bivariate correlation tool to find any connection between the numbers of strains belonging to each resistance phenotype (XDR—extensive drug-resistant; MDR—multidrug resistant; PDR—pan drug resistant) and the numbers of strains with VF or with resistance genes. The categories of resistance were established according to international consensus guidelines [25]. Significant associations were found between the numbers of XDR strains and those with two virulence factors, or those with three virulence genes (Figure 1 and Figure 2). A significant correlation was found between the strains with one VF and those with one resistance gene (Figure 3).

## 4. Discussion

Currently, multidrug resistance in GNB is intensely studied because of the need for improved strategies for the treatment of infection caused by these opportunistic pathogens. In addition, the mechanisms related to the pathogenesis of MDR GNB are intensely being studied in order to gain a better understanding of their adaptability, fitness, and spreading rates [18,21].

Our research focused on the virulence traits of some selected strains belonging to three MDR, XDR, or PDR microorganisms, in relation to their phenotypic profile of AR. All of the investigated bacterial strains, and especially *K. pneumoniae*, showed high resistance levels, in accordance with other studies [30,31]. *K. pneumoniae* strains mostly carried the *bla*_KPC_ carbapenemase encoding gene, followed by a smaller percentage of *bla*_OXA-48_ and *bla*_NDM_ genes. Among the ESBLs, the most prevalent ones were *bla*_CTX-M_*,* followed by *bla*_SHV_ and *bla*_TEM_. For *A. baumannii*, two acquired CHLDs encoding genes were revealed in the analyzed strains: *bla*_OXA-23_ and *bla*_OXA-24_. The same results were found by Gheorghe et al. (2019) in their study [32]. Most of the strains of *P. aeruginosa* were positive for *bla*_IMP_ (Appendix A) and *bla*_VIM_ (Appendix A). 

*P. aeruginosa* is one of the pathogens that harbor numerous VF involved in cell invasion and toxicity. Most of its VF also are involved in biofilm formation. One of the most important VFs is an exopolysaccharide, alginate, with a role of adhesion to the cellular substrate and a protective role against certain environmental adverse conditions. Therefore, its presence is required for the survival of the microorganism and its multiplication within microbial biofilms [33,34,35]. Among VF, the presence of flagella and pili, exotoxin A, protease, alginate production, biofilm formation, and secretion systems III and IV determine the high pathogenicity of *P. aeruginosa* strains [36]. These factors seem not to act alone, but in the presence of other enzymes involved in pathogenicity [37]; these soluble components identified in our study were exoenzymes, such as hemolysins, caseinase, and lecithinase. Apart from gelatinase, we could also find lipase, esculinase, and amylase in strains of *P. aeruginosa*. These virulence factors were also identified in previous studies performed on such strains. Even if amylase, caseinase, and lipase are not literally real virulence factors, but metabolic enzymes (encoded by chromosomal genes), their presence in strains from the *Enterobacterales* and *Pseudomonadaceae* families facilitates adaptability inside the host and even the survival of these bacteria in the external environment, thus increasing the risk of transmission. This is a very important trait among the *P. aeruginosa* strains, which can pass and infect other hosts, thus creating a chain of events that lead to the spread of MDR, with a major impact on the health of the community [38].

*LasA* was described as the first virulence gene found in *P. aeruginosa* with an elastase and protease activity [39], while LasB was described as a metalloenzyme secreted by pathogenic strains of *P. aeruginosa*. A major role of *LasB* is currently designated in bacterial intercellular communication, and this metalloenzyme is also reported in 90% of the urine specimens [37,40,41,42]. A study conducted in Teheran, between June 2013 and June 2014, on pediatric patients with cystic fibrosis, revealed an increased prevalence rate in virulence genes in *P. aeruginosa*, of over 63.1% for *toxA* and 95.4% for *lasB* [43]. Most of the isolated strains in this study had a combination between many virulence genes, especially between *plcH*, *plcN*, *exoS*, *exo U,* and *algD* (Appendix A). The phenotypic expression of these genes consists of the destruction of tissues (*plcH*, *plcN*, *exoS*, and *exoU*) or in the cellular adhesion and formation of biofilms (*algD*) [44]. All these features promote the spreading process and the complex development of biofilms, which, in turn, are substantial factors in the development of multidrug resistance [45]. These intricate relations could be a subject for further study.

The virulence genes are different among the intestinal colonizing *K. pneumoniae* strains, compared with those from other sources [37]. This might be influenced by the fact that several environmental characteristics influence the adherence and survival of these strains, thus leading to the development of complex virulence systems [46]. Emphasizing that *K. pneumoniae* represents one of the most important opportunistic pathogens, this microorganism is frequently involved in urinary and respiratory tract infections, as well as sepsis [47]. The pathogenicity and virulence of *K. pneumoniae* strains are mainly due to the presence of the capsule, with the most common strains being those with the capsule type K1, K2, K5, K54, and K57, as well as K20 (which has two virulence factors, *rmpA* and *wcaG*) [48,49]. It was observed that the virulence gene *wcaG* is involved in the production of the capsular fucose, associated with K1 and K54 strains, while the *rmpA* gene was identified in strains of *Klebsiella* spp. type K2 [50,51,52]. However, the presence of a capsule was not the only contributor to virulence, with several studies underlining the association of various virulence factors in the persistence of this microorganism [53]. In our study, the majority of *K. pneumoniae* strains were positive for more than two virulence factors. 

On the other hand, many VF contribute to a series of mechanisms that have a complex pathogenic response, such as biofilm formation. In this study, we emphasized that most of the encountered VF in all three mainly researched microorganisms were involved in biofilm formation (such as *mrkA*, *ecpRAB*, and *algD*) (Appendix A) and adherence (*fimA*) (Appendix A). We also identified several correlations between the number of virulence genes and antibiotic resistance, depicted as a repetitive pattern in all of the isolation sites. However, the combination between two or more virulence genes was not newly discovered among MDR *K. pneumoniae*. The association between the *ecp* complex and the *fim* system was, for instance, observed in other strains of *K. pneumoniae* isolated from intra-hospital infections [54]. This probably brought a new mechanism of adaptability and biofilm formation for these bacterial strains, in order to increase their survival rate under any circumstances. However, the observed effect might be perilous for the health of the entire community because it might lead to the spread of MDR organisms. In our case, because of the increased number of genes involved in biofilm formation, all correlations need further study because of several interference factors that contribute to the achievement of multidrug-resistant genes/mechanisms. There are studies that emphasize the connection between biofilm formation and antibiotic resistance through various mechanisms [55]. Therefore, this subject needs further analysis.

In bacterial biofilms, tolerance to different classes of antibiotics is due to the different mechanisms involved, such as, the presence of the *ndvB* and *brlR* genes for the aminoglycosides, the efflux pumps for the polymyxin E, and porins (*MexAB-oprM* and *MexCD-oprJ*) for beta-lactams. Antibiotic tolerance is also given by the impermeability of the matrix secreted by the bacterial cells inside the biofilm. In fluoroquinolones, antibiotic tolerance appears because of the decrease in oxygen concentration, the low metabolic activity/dormancy state, or the activity of the efflux pumps [56]. A major part in biofilm formation is also played by the QS complex [39]. Three QS systems are known in *P. aeruginosa*: the most important is LuxI/LuxR, which controls the expression of virulence factors, but a major part is also played by the third non-LuxI/LuxR-type system called the *Pseudomonas quinolone signal system* (PQS). In the first phase, LuxI/LasI synthesizes the 3OC12HSL lactone detected by the cytoplasmic LuxR homologue LasR. The LasR–3OC12HSL complex activates the transcription of target genes, including those encoding VF, such as elastase, proteases, and exotoxin A. One of the targets of LasR–3OC12HSL is lasI; another target is a second luxI homolog called rhlI (synthesizes a second AI, butanoyl homoserine lactone—C4HSL). At high concentrations, this AI binds to RhlR, a second LuxR homolog, RhlR-C4HSL, which activates target genes, including those encoding elastase, proteases, pyocyanin, and siderophores. Among its targets is rhlI, which leads to autoinduction of the second QS system [56]. It is, therefore, already known that biofilms are a result of complex interactions both in *Klebsiella pneumoniae* and in *P. aeruginosa*. 

Another pathogen that is also an important biofilm producer is *A. baumannii* [57], an opportunistic pathogen with major implications in HAIs, responsible for outbreaks and severe infections especially in patients with different immunodeficiencies. Although poor in the production of soluble VF (in our study, almost all of the strains produced only one VF-*ompA*) (see Appendix A), it proved to be an important nosocomial pathogen due to the formation of biofilms (especially on medical devices) and due to its MDR, XDR, or PDR phenotypes [30,32,58,59]. Compared with the soluble VF reported for other Gram-negative bacteria, a small number of these factors have been identified for *A. baumannii*, in comparison with *K. pneumoniae* and *P. aeruginosa*. In a study performed by Zeighami et al. (2019) on a significant number of *A. baumannii* strains, it was observed that most of the strains that carried virulence genes were involved in bacterial biofilm development [60]. Among them, 98% simultaneously showed several virulence genes; the highest prevalence was represented by *csuE* (100%) and *ompA* (81%) genes. The severity of infections produced by pathogenic strains of *A. baumannii* and *P. aeruginosa* affecting both the respiratory system (pulmonary ventilated patients) and the teguments (patients with a compromised epidermal barrier) was due to the presence of QS mechanisms [59,60,61,62]. Gheorghe et al. also reported only strains positives for the *ompA* gene [32]. 

Numerous recent studies carried out in Romania concerning AR had a major impact, both clinically and especially epidemiologically, due to the high spreading capacity of the resistance genes among *Enterobacterales* [63]. One of the most recent studies (Tîrgu-Mureș, 2018) performed on 19 strains of carbapenemase producing *Enterobacterales* showed that the resistance genes detected by PCR were *bla*_KPC_, *bla*_OXA-48_, and *bla*_NDM_ [64]. Gavriliu et al. (2016) mentioned, in a retrospective study conducted in “INBI Matei Balș”, a lower resistance level of *K. pneumoniae* strains isolated from blood cultures, especially to fluoroquinolones, aminoglycosides, and carbapenems, and to third generation cephalosporins, compared with our country’s results reported in EARS-Net [65].

In Romania, Gheorghe et al. (2014), reported for the first time the presence of the *bla*_VIM-4_ gene in *P. aeruginosa* clinical strains (involved in acquired resistance mediated by class I integrons) [66]. A year later, Lixandru et al. described for the first time the presence of the *bla*_VIM-1_ and *bla*_KPC-2_ genes in strains of *K. pneumoniae*, highlighting the urgent need for measures to combat and prevent intrahospital infections [67]. Two years later, Georgescu et al. (2016), mentioned for the first time the presence of the *bla*_OXA-72_ gene in *A. baumannii* strains, responsible for plasmid-mediated resistance through plasmid pAB120 [68].

Only a few papers have been published in Romania on the topic of antibiotic resistance and virulence; for instance, Delcaru et al. (2017) investigated these associated features in UTI strains; their study revealed a high ability of the UTI strains to adhere to HEp-2 cells. The Gram-negative UTI strains produced pore-forming toxins (hemolysins, lecithinase, and lipase), proteases, amylase, and siderophore-like molecules resulted from the process of esculin hydrolysis. In their opinion, searching for the correlation between resistance and virulence factors could provide information useful for anticipation of the clinical evolution and risk of UTI recurrence [22].

Our study depicted numerous carbapenem and ESBL-encoding genes, especially in *K. pneumoniae*, most of them being encoded by β-lactamases of class A (KPC, TEM, SHV), B (NDM) and D (OXA-48). This complex mixture was reflected in the resistance profile of the identified strains, most of them being classified as either PDR or XDR. These strains also had several VF, most of them being involved in biofilm formation or tissue destruction. However, despite very careful selection of the analyzed strains, the major limitation of our study was the reduced number of isolates (being selected strains with epidemiologically important resistance profiles), which might influence the statistical analyses.

Studies similar to this one can help gain a better understanding of the adaptability of a pathogen to the host environment. Finally, a good adaptation is as a result of a well-balanced distribution of energy resources necessary for growth, multiplication, and colonization; virulence factors expressed in a cell-density-dependent manner; and the pathogen’s survival in opposition with innate and adaptive immune system effectors, eventually with antibiotics too. This network of processes modulate the reproductive fitness of the pathogen and influence the community-acquired infections and HAIs evolution; therefore, gaining a better understanding of them will help with the development and implementation of appropriate prevention, diagnosis, and treatment actions [69]. 

## 5. Conclusions

Our study concerning the resistance and virulence profile of carefully selected MDR, XDR, and PDR strains of Gram-negative bacilli from the ESKAPE group (*Klebsiella pneumoniae*, *Acinetobacter baumannii,* and *Pseudomonas aeruginosa*) revealed a high repertoire of ARGs and VMs. The most frequently expressed virulence factors were those involved in host cell/tissue destruction and biofilm formation. Such phenotypes are of interest due to their fitness, adaptability, and capacity of dissemination, both in the environment and in the host. This study, which focused on the genotype−phenotype association of resistance and virulence (less approached in Romania), could contribute to a better understanding of bacterial pathogenesis in patients from intensive care units (where biofilm formers and antibiotic-resistant strains spread). The results of this study show that it is imperative to continue monitoring antibiotic resistance, which is often associated with virulence factors. The careful surveillance of both the environmental and host conditions in which the bacteria/microorganisms localize themselves, grow, and colonize is also of great importance because it helps in establishing the pathogenic characteristics for a better prevention and treatment policy regarding healthcare-associated infections.

## Figures and Tables

**Figure 1 pathogens-12-00609-f001:**
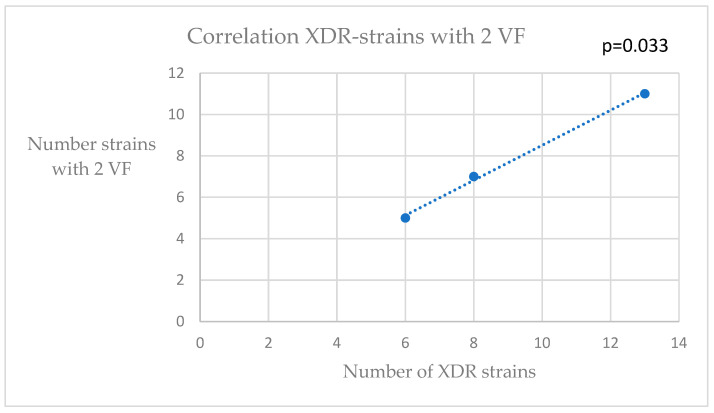
Correlation XDR—strains with two VF (virulence factors).

**Figure 2 pathogens-12-00609-f002:**
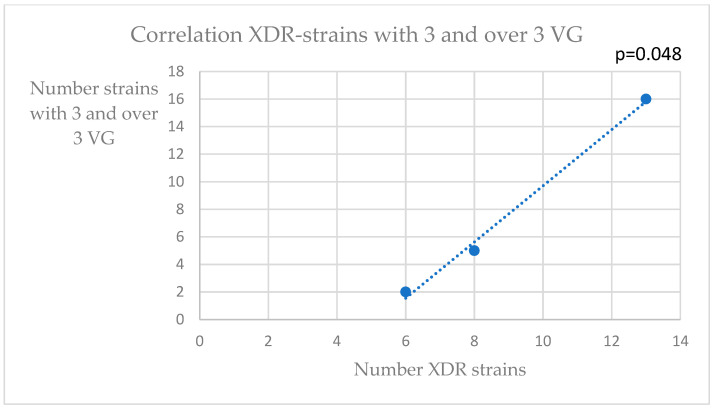
Correlation XDR—strains with three or more VG (virulence genes).

**Figure 3 pathogens-12-00609-f003:**
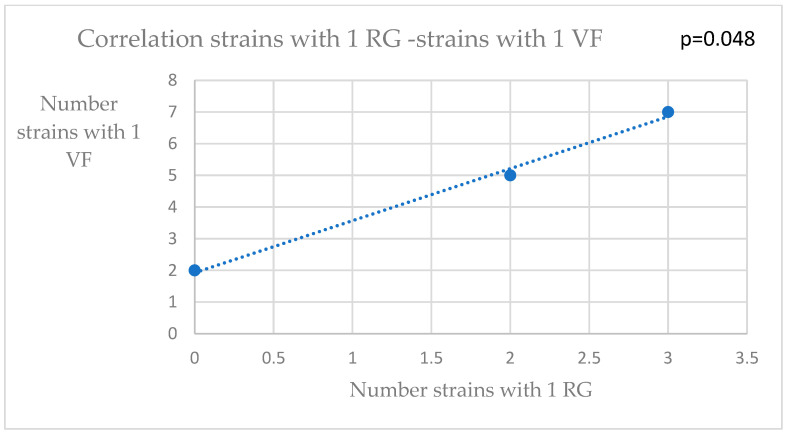
Correlation strains with one RG (resistance gene)—strains with one VF (virulence factor).

**Table 1 pathogens-12-00609-t001:** The primers used for the detection of virulence and antibiotic resistance genes.

Target Gene	Primer Name	Primer Sequence	Amplicon Size (bp)
**Primers used for antibiotic resistance genes detection**			
*bla* _IMP_	IMP-F IMP-R	5′-GGAATAGAGTGGCTTAAYTCTC-3′ 5′-GGTTTAAYAAAACAACCACC-3′	232
*bla* _VIM_	VIM-F VIM-R	5′-GATGGTGTTTGGTCGCATA-3′ 5′-CGAATGCGCAGCACCAG-3′	390
*bla* _OXA-48_	OXA-48-F OXA-48-R	5′-ATGCGTGTATTAGCCTTATC-3′ 5′-CTAGGGAATAATTTTTTCCT-3′	438
*bla* _NDM_	NDM-F NDM-R	5′-GGGCAGTCGCTTCCAACGGT-3′ 5′-GTAGTGCTCAGTGTCGGCAT-3′	621
*bla* _KPC_	KPC-F KPC-R	5′-GCTCAGGCGCAACTGTAA-3′ 5′-AGCACAGCGGCAGCAAGAAAG-3′	798
*bla* _CTX-M_	CTX-M-F CTX-M-R	5′-CGCTGTTGTTAGGAAGTGTG-3′ 5′-GGCTGGGTGAAGTAAGTGAC-3′	754
*bla* _TEM_	TEM-F TEM-R	5′-ATGAGTATTCAACATTTCCGT-3′ 5′-TTACCAATGCTTAATCAGTGA-3′	1080
*bla* _OXA-23_	OXA-23-F OXA-23-R	5′-ATGAGTTATCTATTTTTGTC-3′ 5′-TGTCAAGCTCTTAAATAATA-3′	501
*bla* _OXA-24_	OXA24/40-F OXA24/40-R	5′ GCAGAAAGAAGTAAARCGGGT3′ 5′ CCAACCWGTCAACCAACCTA3′	270
*bla* _OXA-51_	OXA-51-F OXA-51-R	5′-TAATGCTTTGATCGGCCTTG-3′ 5′-TGGATTGCACTTCATCTTGG-3′	353
**Primers used for virulence markers detection**			
*epsA*	epsA-F epsA-R	5′-AGCAAGTGGTTATCCAATCG-3′ 5′-T ACCAGACTCACCCATTACA-3′	451
*ompA*	ompA-F ompA-R	5′-CGCTTCTGCTGGTGCTGAAT-3′ 5′-CGTGCAGTAGCGTTAGGGTA-3′	531
*plcH*	plcH-F plcH-R	5′GAAGCCATGGGCTACTTCAA-3′ 5′AGAGTGACGAGGGGTAG-3′	466
*plcN*	plcN-F plcN-R	5′-GTTATCGCAACCAGCCCTAC-3′ 5′-AGGTCGAACACCTGGAACAC-3′	307
*exoU*	exoU-F exoU-R	5′-CCGTTGTGGTGCCGTTGAAG-3′ 5′-CCAGATGTTCACCGACTCG-3′	134
*exoT*	ExoT-F ExoT-R	5′-AATCGCCGTCCAACTGCATGCG-3′ 5′-TGTTCGCCGAGGTACTGCTC-3′	152
*exoS*	ExoS-F ExoS-R	5′-ATCGCTTCAGCAGAGTCCGTC-3′ 5′-CAGGCCAGATCAAGGCCGCGC-3′	1352
*algD*	algD-F algD-R	5′-ATGCGAATCAGCATCTTTGGT-3′ 5′-CTACCAGCAGATGCCCTCGGC-3′	1310
*TC*	TCF TCR	5′-TATTTCGCCGACTCCCTGTA-3′ 5′-GAATAGACGCCGCTGAAATC3′	752
*fimH*	fimH-F fimH-R	5′-TGCAGAACGGATAAGCCGTGG3′ 5′-GCAGTCACCTGCCCTCCGGTA3′	506
*fimA*	fimA-F fimA-R	5′-CGGACGGTACGCTGTATTTT-3′ 5′-GCTTCGGCGTTGTCTTTATC-3′	500
*hlyD*	hlyD-F hlyD-R	5′-CTCCGGTACGTGAAAAGGAC3′ 5′-GCCCTGATTACTGAAGCCTG3′	904
*hlyA*	hlyA-F hlyA-R	5′-AACAAGGATAAGCACTGTTCTGGCT-3′ 5′-ACCATATAAGCGGTCATTCCCGTCA-3′	1177
*cnf1*	cnf1-F cnf1-R	5′-GAACTTATTAAGGATAGT-3′ 5′-CATTATTTATAACGCTG-3′	544
*ecpRAB*	ecpRAB-F ecpRAB-R	5′-CCTATGTAATTAATGGCAGGTTT-3′ 5′-GCTGTTCATAAAGGATGAAATATC-3′	1025
*mrkA*	mrkA-F mrkA-R	5′-CGGTAAAGTTACCGACGTATCTTGTACTG-3′ 5′-GCTGTTAACCACACCGGTGGTAAC-3′	498
*mrkD*	mrkD-F mrkD-R	5′-CTGACGCTTTTTATTGGCTTAATGGCGC-3′ 5-′GCAGAATTTCCGGTCTTTTCGTTTAGTAG-3′	756

**Table 2 pathogens-12-00609-t002:** The distribution of Gram-negative strains selected for this study.

Isolation Source	Number (n) of Isolated Species		
	*K. pneumoniae*	*Acinetobacter* spp.	*P. aeruginosa*
Urine	n = 28	n = 3	n = 4
Blood cultures	n = 10	n = 3	n = 4
Others *	n = 13	n = 5	n = 2

* Biological fluids, wounds, endotracheal secretions, stool cultures, and rectal swabs.

**Table 3 pathogens-12-00609-t003:** Bacterial strains producing soluble enzymatic factors.

Microorganism	Virulence Factors	Urine N (%) *	Blood Culture N (%) *	Bile Culture N (%) *	Wound N (%) *	Bronchial Aspirate N (%) *	Rectal Swab N (%) *	Drainage Tube N (%) *	Ascitic Fluid N (%) *	Stool Culture N (%) *	*p*-Value **
*K. pneumoniae*	amylase	9 (34.6)	1 (10)	0	0	0	1 (50)	1 (50)	1 (100)	0	0.349
	esculinase	25 (96.2)	10 (100)	1 (100)	1 (100)	4 (100)	2 (100)	2 (100)	1 (100)	2 (66.7)	0.556
	lecithinase	2 (7.7)	2 (20)	0	0	0	0	0	0	2 (66.7)	0.251
	caseinase	0	2 (20)	0	0	1 (25)	0	1 (50)	0	1 (33.3)	0.047
	lipase	9 (34.6)	3 (30)	0	0	1 (25)	0	1 (50)	0	2 (66.7)	0.833
	gelatinase	1 (3.8)	3 (30)	0	0	0	0	0	1 (100)	0	0.059
	hemolysin	2 (7.7)	0	0	0	0	0	0	0	0	0.991
*P.* *aeruginosa*	protease	2 (50)	2 (50)	0	1 (50)	0	0	0	0	0	NA
	amylase	0	0	0	1 (50)	0	0	0	0	0	0.108
	esculinase	0	1 (25)	0	0	0	0	0	0	0	0.435
	lecithinase	3 (75)	4 (100)	0	2 (100)	0	0	0	0	0	0.435
	caseinase	1 (25)	2 (50)	0	1 (50)	0	0	0	0	0	0.732
	lipase	0	2 (50)	0	0	0	0	0	0	0	0.153
	hemolysin	4 (100)	4 (100)	0	2 (100)	0	1 (100)	0	0	0	NA
*Acinetobacter* spp.	amylase	0	0	0	0	0	0	1 (100)	0	0	0.088
	lipase	0	1 (33.3)	0	0	1 (100)	1 (100)	0	0	0	0.266

* N = number of isolates with a virulence factor. Percentages were calculated considering the total number of samples of a specific strain encountered in each isolation source. ** *p*-values reflected the significance between the differences in distribution of a virulence gene or resistance gene over the source types. Every *p*-value was obtained using a chi-test and only those values < 0.05 were considered significant. NA = not applicable.

**Table 4 pathogens-12-00609-t004:** VF patterns encountered in the studied strains.

Microorganism	Virulence Gene Pattern	Urine N (%) *	Blood Culture N (%) *	Other N (%) *	*p*-Value **
*K. pneumoniae*	*mrkA*	0	1 (10)	2 (13.3)	0.11
	*fimA*	0	1 (10)	0	0.19
	*fimA, mrkA*	0	2 (20)	0	0.03
	*fimA, ecpRAB*	0	1 (10)	0	0.19
	*ecpRAB, mrkA*	8 (30)	2 (20)	8 (53.3)	0.22
	*fimA, ecpRAB, mrkA*	10 (38)	2 (20)	4 (26.6)	0.56
	*fimH, ecpRAB, cnf1*	1 (3.8)	0	0	0.99
	*fimH, mrkA, cnf1*	1 (3.8)	0	0	0.99
	*fimH, fimA, ecpRAB, mrkA*	1 (3.8)	0	0	0.99
	*fimA, hlyD. hlyA, ecpRAB*	1 (3.8)	0	0	0.99
	*hlyD, hlyA, ecpRAB, mrkA*	0	1 (10)	1 (6.66)	0.23
	*fimH, hlyD, hlyA, ecpRAB, cnf1*	1 (3.8)	0	0	0.99
	*fimH, fimA, ecpRAB, mrkA, cnf1*	1 (3.8)	0	0	0.99
*P. aeruginosa*	*plcH, plcN, exoU, algD*	0	1 (33.3)	0	0.9
	*plcH, plcN, exoU, exoS, algD*	2 (50)	1 (33.3)	0	0.9
	*plcH, plcN, exoT, exoS, algD*	1 (25)	1 (33.3)	1 (50)	1
	*plcH, plcN, exoU, exoT, exoS, algD*	0	1 (33.3)	0	0.9
*Acinetobacter* spp.	*ompA*	3 (100)	3 (100)	4 (80)	1

* N = number of isolates with a virulence gene or an association of virulence genes. Percentages were calculated considering the total number of samples of a specific gene or association of genes encountered in each isolation source. Other includes biological fluids, wounds, endotracheal secretions, stool cultures, and rectal swabs. ** *p*-values reflected the significance between the differences in distribution of a virulence gene or resistance gene over the source types. Every *p*-value was obtained using a Fisher-test and only those values < 0.05 were considered significant.

## Data Availability

Not applicable.

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
