# Peer review of "Snapshot of Phenotypic and Molecular Virulence and Resistance Profiles in Multidrug-Resistant Strains Isolated in a Tertiary Hospital in Romania"

_pathogens, 2023, doi:10.3390/pathogens12040609_

Round 1

Reviewer 1 Report (Previous Reviewer 2)

Line 30:  "Insights" not insides.  Line 47:  "coma" after (antibiotic resistance)  Line 48:  "Consequences"  rather than different rhythms.  Line 60-61  virulence genes in MDR GNB "is as equally important as the phenotypic ...... .

Line 84: mechanism or mechanisms ?  Line 90 "other newly discovered proteins"  Vague; add references or eliminate.  Line 101  "a series of studies have emphasized ......  "references should be added, otherwise the sentence is .  Line 103-104  "epidemiological treat"  ?  Line 179  "Table 1" rather than  table 1.  Line 328:  "environmental"  rather than environment. 

Line 211 " Table 1", rather than table 1

The sequence of Tables may be confusing  Table 1,  then, Table 2, Then Supplementary Table 1, then, Table 3, the, n Supplementary Table 2, etc. This can be  confusing. Explain the relevance to the readers.

Author Response

Esteemed Reviewer,

Thank you very much for all your valuable suggestions that have greatly helped us to improve the manuscript. We tried to answer to your pertinent and useful observations. We marked our changes using Track Changes throughout the manuscript. Our responses are below (they have been marked using gray highlight).

Reviewer 1

  1. Line 30: "Insights" not insides.  Line 47:  "coma" after (antibiotic resistance) Line 48:  "Consequences"  rather than different rhythms.  Line 60-61 virulence genes in MDR GNB "is as equally important as the phenotypic ...... .

Answer: Thank you very much for this suggestion, in the revised version of the manuscript we corrected all the above-mentioned aspects (pages 1-2, lines 30; 47; 49; 60-61).

  1. Line 84: mechanism or mechanisms ? Line 90 "other newly discovered proteins"  Vague; add references or eliminate.  Line 101  "a series of studies have emphasized ......  "references should be added, otherwise the sentence is .  Line 103-104  "epidemiological treat"  ?  Line 179  "Table 1" rather than  table 1.  Line 328:  "environmental"  rather than environment.

Answer: These problems have been addressed (page 2; 3; 4; 10, lines 84; 91; 104; 105; 181; 323);

  1. Line 211 " Table 1", rather than table 1

Answer: Thank you very much for this observation, we performed the necessary correction (page 6, line 199).

  1. The sequence of Tables may be confusing Table 1, then, Table 2, Then Supplementary Table 1, then, Table 3, the, n Supplementary Table 2, etc. This can be Explain the relevance to the readers.

Answer: Thank you for this observation, because the investigated strains were very heterogeneous they were firstly analyzed according to their distribution by the sources of isolation (Table 2), then after the antibiotic resistance profiles (Supplementary Table 1), the categories of enzymatic virulence factors (Table 3), the virulence and antibiotic resistance genes (Supplementary Table 2) and after the association between some VM and specific isolation sources in order to reveal significant correlations. Also, we moved the Supplementary Table 1 and Supplementary Table 2 in the “Supplementary materials” chapter (at the end of article, pages 24-25).

Reviewer 2 Report (Previous Reviewer 3)

The manuscript was greatly improved. 

Nonetheless, a few things still have to be addressed:

- Line 196: suggested correction: “…strains of K. pneumonia, P. aeruginosa and Acinetobacter spp.”

- Concern to be addressed: The p-values shown in tables 3, 4 and Supplementary Table 2 are not clear to me. It is stated that “*P-values reflected the correlation between a virulence gene or resistance gene and the source types”, however, most of the times there are several sources (urine, blood, wound, etc.) from where the same gene was retrieved, but there is only one p-value. Please clarify. 

-        Supplementary figures must be provided regarding the images of the gels after electrophoresis. The positive controls must be pinpointed. 

Author Response

Esteemed Reviewer,

Thank you very much for all your valuable suggestions that have greatly helped us to improve the manuscript. We tried to answer to your pertinent and useful observations. We marked our changes using Track Changes throughout the manuscript. Our responses are below (they have been marked using gray highlight).

The manuscript was greatly improved. Nonetheless, a few things still have to be addressed:

  1. Line 196: suggested correction: “…strains of K. pneumonia, P. aeruginosa and Acinetobacter spp.”

Answer: Thank you very much for the suggestion, in the revised version of the manuscript this has been corrected (page 6, lines 198).

  1. Concern to be addressed: The p-values shown in tables 3, 4 and Supplementary Table 2 are not clear to me. It is stated that “*P-values reflected the correlation between a virulence gene or resistance gene and the source types”, however, most of the times there are several sources (urine, blood, wound, etc.) from where the same gene was retrieved, but there is only one p-value. Please clarify.

Answer: We have corrected each sentence from tables 3, 4 and Supplementary Table 2, in order to provide a more specific explanation of each p-value: “P-values reflected the significance between the differences in distribution of a virulence gene or resistance gene over the source types.”

  1. Supplementary figures must be provided regarding the images of the gels after electrophoresis. The positive controls must be pinpointed.

Answer: Thank you for this observation, we included supplementary figures containing the images of gel electrophoresis for selected amplicons obtained during PCR reactions performed for β-lactam resistance genes and for virulence markers; the positive control used in each PCR reaction was marked in each case (pages 19-23). All the figures are listed in the “Supplementary materials” chapter (at the end of article and cited throughout the text – page 6, 10, 11 lines 214; 217; 285; 286; 315; 316; 339; 340; 380).

Additionally, we have made other changes:

  • We italicized all the ARGs and virulence markers throughout the manuscript;
  • We added positive strains for virulence and β-lactamase encoding antibiotic resistance genes.
  • We replaced coworkers with ,,et ’’ (page 11, 12 lines 384 and 392).
  • We replaced “P-values reflected the correlation between a virulence factor and the source types” with “P-values reflected the significance between the differences in distribution of a virulence gene or resistance gene over the source types” (Table 3 and Supplementary Table - page 8, and Table 4- page 9).
  • We have also rephrased the sentence “The association of genes fimA and mrkA was more likely to be found in blood cultures than in other isolation sources” and we have explained the obtained result: “The association of fimA and mrkA genes was more likely to have a source-related distribution.” (page 7, lines 242-243).
  • All the tables were cited correctly throughout the manuscript: Table 1, Table 2, Table 3, Table 4, Supplementary Table 1, and Supplementary Table 2.

This manuscript is a resubmission of an earlier submission. The following is a list of the peer review reports and author responses from that submission.

Round 1

Reviewer 1 Report

In this study, antimicrobial resistance and virulence factors of several Gram-negative bacilli isolated in a Romanian hospital have been investigated by both phenotypic and genetic methods. However, the study has some limitations, including a very low number of isolates selected for the virulence factor analysis. Therefore, some of the conclusions raised from these findings could be biased.

There is also missing information in the Materials and Methods section, and results obtained should be further analyzed.

Writing should be revised too.

Specific comments:

Some parts of the Materials and Methods section are not clearly described and should be specified. For example, there is missing information about the source of the isolates (types of sample), the total number of isolates of each bacterial species included in the phenotypic antimicrobial susceptibility test and the antimicrobials / antimicrobial classes used in the evaluation of their antibiotic resistance.

Also, no references are provided for the primers used in the detection of resistance and virulence-related genes. Since different DNA fragments were amplified, were all PCR reactions (for resistance and virulence genes, respectively) carried out at the same conditions as stated in the manuscript?

It is confusing which virulence factors were amplified in the PCR study, since they are mentioned in lines 173-178 but also different ones are described later in lines 187-190. Regarding beta-lactamase genes, some of the genes mentioned in table 1 (Results section) are not included in the methodology section, i.e. OXA-51, 23, 24.

No positive and negative controls have been included in any of the tests performed.

The results obtained in the study should have been analyzed with more detail and described more clearly, including information from all the bacterial species identified. It could have been divided into subsections.

Please, replace “100% resistant” by “resistant”.

In table 1, “Resistant genes” should be replaced by “Resistance genes” or “Beta-lactamase encoding genes”. Also, according to this table, there are 12 K. pneumonia strains, but a total of 13 strains are described in column 4.

In lines 248-250 it is stated that “Strains with none or less than two virulence genes were likely to come from urine”, while in the abstract it is said the contrary: “A statistical relevant association was found between the number of virulence genes and the source of isolation, the strains involved in urinary tract infections (UTI) being the most virulent”.

Reviewer 2 Report

A well written manuscript addressing "Multidrug Resistant Strains" of Gram negative bacilli. Death from these bacilli is increasing, and therefore treatment requires much diligence and care.

Strengths of the manuscript:  (1) Although there are several (many) strains, the authors concentrated on basically three (page3, line112) and described their actions, and the changes  in the severity of the progress of disease(s) clearly. (2) They provided the virulence factors responsible for eventual death, e.g. pili, efflux pumps, intrinsic resistance to therapy, toxins, and others. (3) They searched for drugs that would lesson multi drug burden  resistance. (4) they identified the various genes that contribute to drug resistance.

The enzymatic factors produced by bacterial strains are clearly depicted in supplemental Table 1. Figure 2 also depicts the production of hemolysis.(4) Table 1, and Table 2, clearly represent the virulence genes and resistant strains observed.

The references presented offer a span of older and new literature. 

In all, the manuscript presents a clear understanding and documentation  of the pathways that lead to  multi drug resistance, at least from older to current advances.

Some suggestions:

 Page 2 line 45 - eliminate (etc).  

Some sentences are too long, example page 11 line 401-405 is one  sentence. It is too long and can be distracting. About 3-4 lines for a sentence is sufficient (my opinion). I believe the authors can accomplish this task.

What I would very much like to see added in more detail is a description of   the impact of this manuscript on the field of Multidrug resistance. "This has been eluded to in line 28-31, and also in lines 408-413". I believe it would be of great benefit to other authors, be they physicians or scientists (or both). A separate paragraph describing the relevance and impact of the results of this work would be addition.     

Reviewer 3 Report

The manuscript entitled “Snapshot of Phenotypic and Molecular Virulence and resistance Profiles in Multidrug Resistant Strains Isolated in a Tertiary Hospital in Romania” does not reach, in my opinion, the quality and scientific standards needed for publication.

Methods are poorly described. 148 isolates from various biological samples from hospitalized patients from units with high epidemiological risk were included in this study, thus, as expected they were multidrug-resistant strains harboring virulence genes as well. My major concern is that this study does not meet the requirement of scientific soundness and rigor to be published.  

However, below, I pinpointed a few comments and suggestions showing my concerns that may be useful for authors for future improvement:

-        Lines 47-50: what is in brackets does not flow with the rest of the sentence. It is confusing as it is.

-        The introduction section lacks information regarding an important mechanism of resistance, that is the production of beta-lactamases, which is quite relevant in GNB. Only the mechanisms regarding the modified permeability of the outer membrane and the efflux pumps are pinpointed.

-        Lines 133-137: There are no references to support the methods used.

-        Line 142: “using previously described primers”, yet the reference is missing.

-        Line 154: Correct as follows: “specific substrates for bacterial toxins/enzymes.”

-        Lines 162-163: Correct as follows: “The protease activity of caseinase and gelatinase was determined using 15% soluble casein agar and 3% gelatin as substrate, respectively.”

-        Lines 174-194: this section is not clear. Authors say “We used genes involved in biofilm formation (algD, ecpRAB, mrkA, mrkD, ompA, epsA), in adherence (TC, fimH, fimA), in tissue distruction (plcH, plcN) and in toxin production (cnfI, hlyA, hlyD, exs complex).” But then in the same section they also say “Genomic DNA was used as a template for the PCR screening of 7 virulence genes encoding for protease IV, three exoenzymes – exoS, exoT, exoU, two phospholipases - plcH (hemolytic phospholipase C) and plcN (non-hemolytic phospholipase C) and for the alginate.” Moreover, the primers used have not been mentioned or proper citations were not made.

-        Lines 200-201: This sentence is very vague and expresses an expected outcome regardless of the results of this study since the strains used were isolated from various biological samples from hospitalized patients from units with high epidemiological risk.

-        Lines 202-214: These results may be better organized in a table and instead of the percentages, the number of isolates presenting the respective result out of the total should be included.

-        Lines 220-221: “For A. baumannii, blaOXA-51 were present in all analyzed strains (100%)”. The presence of oxa-51 is normal since this gene is intrinsic to A. baumannii species.

-        Line 224: Supplemental figure 1?? Why supplemental?

-        Lines 238 and lines 240: Both called supplemented figures are needless. They do not represent a concrete result.

-        Line 234: This table is quite useless since the bacterial strains/isolates are not specified. So, what is the point of the first column of the table. Only species names are referred and repeated… It does not harbor any relevant information.

-        Lines 248-252: “Strains with none or less than two virulence genes were likely to come from urine (p=0.0128).” This result is vaguely presented and explained. Moreover, Table 2 does not harbor any relevant information…

-        The discussion section seems more an introduction section… Many studies are referenced randomly.

-        There are some typos and grammar errors throughout the manuscript that need revision. Also, genes’ names must be italicized.